# Urban Land Use Survey Methods: A Discussion on Their Evolution

Ioannis A. Pissourios 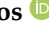

School of Architecture, Land and Environmental Sciences, Neapolis University Pafos, 2 Danais Avenue, 8042 Pafos, Cyprus; i.pissourios@nup.ac.cy

**Abstract:** Although the tradition of surveying and analyzing urban land uses for town planning purposes dates back to the 19th century, the evolution of survey methods has not been studied in detail. With the intention of filling this gap, the present article reviews the pertinent Anglo-American literature on survey methods, published from the beginning of the 20th century to date, and highlights the key contributions. Additionally, it proposes a periodization of the methodological evolution in three phases and identifies the main discussions developed on survey methodology, so as to provide a basis for more structured research on the subject matter.

**Keywords:** land use; urban use; survey methodology; fieldwork; inventory

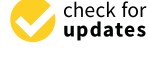



## 1. Introduction

This article reviews the Anglo-American literature pertinent to the survey methods of urban land uses. The term 'survey' is understood, within this article, as a targeted, coordinated, and systematic inventory process that involves the collection of primary data (i.e., fieldwork) and yields records of existing land uses in the form of a database and/or a map. Figure 1 provides a snapshot of this inventory procedure through presenting the field listing form used in a survey of land uses. Maps such as those in Figures 2 and 3 illustrate the main or even the single output of a survey of urban land uses. The term 'urban land use' is used to define the organizational unit that occupies identifiable space at a fixed location and performs a specific urban function [1] (p. 31), [2] (pp. 221–226), [3] (p. 274). Except when special reference is made, no differentiation is made in this article between 'urban use' and 'urban land use' [4]. The former term indicates each individual activity occupying a spatial location (see Figure 2), while the latter indicates the *predominant* activity within a defined spatial unit (see Figure 3).

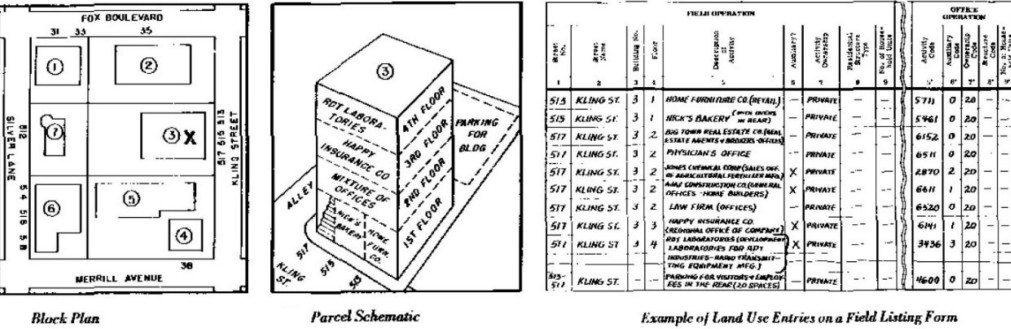

**Figure 1.** Example of a field listing form for the survey of urban uses [5] (p. 21).

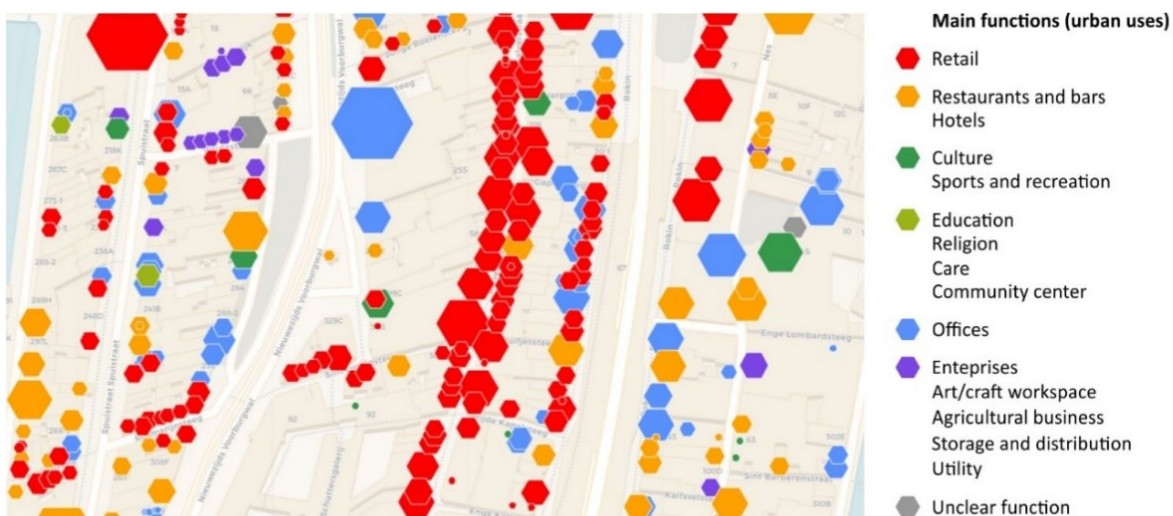

**Figure 2.** Digital database of Amsterdam representing existing urban uses. Urban uses are symbolized as hexagons, with the size of each symbol being proportional to the surface area of the corresponding use. Source: City of Amsterdam. Available at https://maps.amsterdam.nl/functiekaart/?LANG=en (accessed on 4 February 2022).

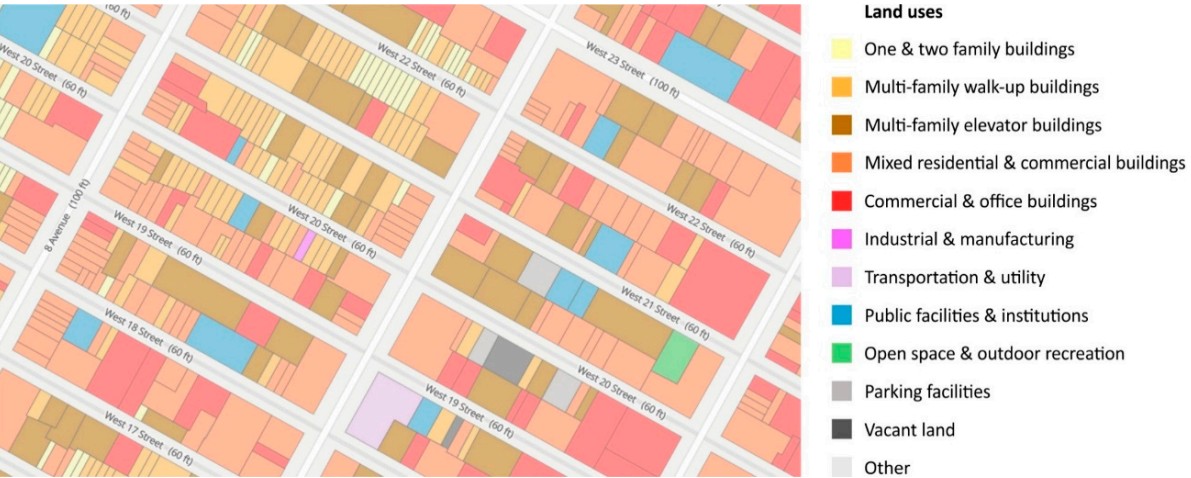

**Figure 3.** Digital database of New York, representing the existing urban land uses per plot. Source: NYC Planning. Available at https://zola.planning.nyc.gov/about/ (accessed on 4 February 2022).

The stimulus for this literature review was a recent paper, which revealed that contemporary planning literature does not provide a comprehensive list of the available methods for the survey of urban land uses [6]. In search of the reasons for this omission, the author of this publication explored three possible explanations: (a) survey methods of urban land uses are already well-developed and there is no room for further development, (b) the analysis and planning of urban space no longer requires land use surveys, and (c) remote sensing and crowdsourced geo-platforms can provide a clear picture of the distribution and patterns of urban land use in space, eliminating the need for field surveys. The author, however, concluded that none of the above hold true. Even more, he stressed that the identified omission negatively affects the potential of urban analysis, as a sound methodological framework would improve the quality of collected data. Furthermore, such a survey would reduce the time and resources needed for the survey, allow for a wider range of analytical options, facilitate future survey updates, and simplify the conduct of comparative studies.

The finding that the current literature is insufficient regarding the survey of urban land uses can be considered unexpected for two reasons. First, because contemporary planning

practice is mainly based on land use planning [7–11], and this was more or less the case throughout the last century [11]. In other words, established practices and frameworks for the inventory, analysis, and planning of urban land uses should already be in place. Second, because all planning begins with and resorts to some sort of analytical work; thus, existing land uses comprise the essential basis for their planning. Specifically, their survey and mapping allows planners to answer important questions relating to how land is currently used, what the dynamics of land use changes are, what land use changes can be made in accordance with a set of rules, and how these changes impact land use [3] (p. 274). Due to the above, a planner would reasonably expect to easily find well-developed methods for the survey of urban land uses in the current literature. Each method would be accompanied by tested solutions at the technical level and discussions of their relation to the most recent technological developments. In short, one would expect to find a mature methodological framework—an expectation, however, that is not met. Additionally, from an educational standpoint, it is difficult to provide under- or post-graduate students in architecture or planning studies with a contemporary reading that elaborates on the basic theoretical underpinnings and explains how the survey of urban land uses is conducted.

Given the key position of land use surveys in planning practice, scholars have shown little interest in studying the historical development of urban land use survey methods. One exception is Robert A. Clark's book: *Selected References on Land Use Inventory Methods*, which was published in 1969 and therefore misses the developments of the last 50 years [12]. Studies that focus on the evolution of planning methodology (see: [13–16]) do include research on urban land use surveys but do not cover the specific subject matter in further detail. Studies on the ways land use maps have evolved (see: [17–21]) can be regarded as closely affiliated; however, they concentrate on the cartographical aspect, rather than on survey methods, and show no particular interest in urban space.

In light of the above research gap, this article performs an in-depth review of the Anglo-American literature pertinent to the survey methods of urban land uses. In addition, it examines selected field surveys and considers wider developments in planning theory and methodology, when these provide critical understandings of the way(s) that survey methods have evolved or have been transformed. The objective of this study is threefold. First, *to highlight key contributions*, given that some past publications on the subject matter remain exceptionally advanced even by today's standards. Second, to *present the main elements that constitute a land use survey method* and *to discuss their evolution over time*. Finally, *to provide a basis for further discussions and research on survey methods*, because their current state impacts the potentialities of urban analysis negatively, and because methodological and conceptual histories are the most neglected types among planning histories [22].

With respect to the period covered, the study is limited to the relevant literature published in the 20th century onwards. This is a choice initially dictated by practical reasons, as older literature on the subject matter is scarce and difficult to access. Moreover, the simplicity of older surveys, evident in the resultant figure-ground maps with labels indicating important uses (see Figure 4), limits room for discussion of their methodological background This by no means overlooks or underestimates the nodal importance of these endeavors in cartography and in the advancement of survey techniques (i.e., measurement and representation) of the built environment [23]. Indeed, the role of these attempts in the formation of land use survey methods for the period studied herein comprises a solid research topic, albeit one falling outside the scope of the present study.

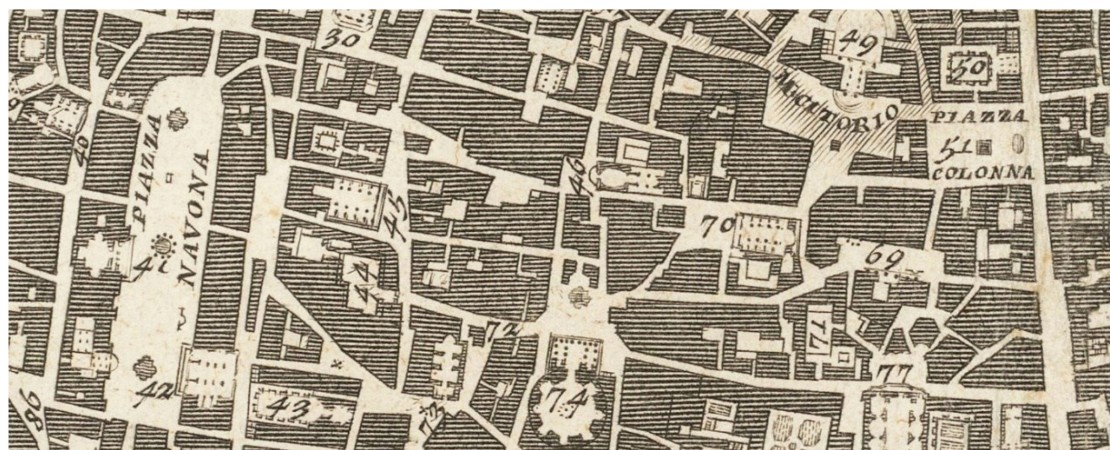

**Figure 4.** Detail from Giambattista Nolli's '*Nuova Pianta di Roma*', published in 12 sheets in 1748. The numbers on the map indicate important uses, which are named on the border of each sheet. This ichnographic (figure-ground) mapping approach was widely used until the early 20th century, when technical advances allowed the fuller use of color printing and made a wider range of thematic detail possible. Figure-ground or ichnographic maps are prepared to illustrate the relationship between built and unbuilt space. Land coverage of buildings is visualized as a solid mass (figure), while public spaces formed by streets, parks, and plazas are represented as voids (ground). The technique was rediscovered and revived by the architectural theorist Colin Rowe in the late 1950s [23].

## 2. The Early Practice-Led Foundations in Survey Methods

The tradition of surveying urban space prior to any plan formulation is over a century old. At the end of the 19th century, Patrick Geddes applied the fundamental '*survey-before-plan*' methodological approach [15,16,24], arguing that '*civic surveys*' were an essential prerequisite to any exercise in town planning [25] (pp. 355–358). The latter comprised comprehensive surveys of social, cultural, and environmental conditions and their historical antecedents, and did not specifically include land use surveys. Geddes' approach was originally applied in the survey of the old town of Edinburgh in 1900 and soon came to be recognized as a key element in urban planning methodology with a lasting impact upon the latter [13].

Geddes' method was supported in the years to come by both practitioners and academics in the field of urban planning. Raymond Unwin, the leading planning practitioner in England at the time, supported the importance of surveys, even though he questioned the practicality of carrying out such extended inventories. Unwin also prompted planners to survey the distribution of residential, business, and manufacturing areas; the distribution of parks, public and other open spaces; and the extent of each [26] (pp. 140–141). Likewise, Patrick Abercrombie, one of the most active planning practitioners of his time and editor of the journal Town Planning Review [21], widely supported the ideas of Geddes [27] (pp. 85–86), as well as the preparation of '*surface utilization plans*'. These were plans in which "the existing use of every square yard of ground is shown" [28] (p. 187). In his book, Abercrombie [29] also offered direct guidance on the technical part of a survey in a practical instructive sense [14] (p. 123), which underlines the importance he attributed to the connection between theory and practice.

Large-scale land use surveys, which were organized in England and in the United States to act as work-relief programs, played a crucial role in the development of survey methods. In England, the *Civic Survey of Greater London* program, launched in 1915 and continued until 1919, was an initiative to prevent and relieve distress caused by the outbreak of World War I and the consequent rise in unemployment among architects and surveyors [21]. This is one of the first systematic large-scale surveys of land uses in the contemporary history of urban studies. (In relation to other urban surveys in the beginning of the 20th century, see: [16] as well as [17–20,30]. On the history of land use surveys even

before the 20th century, see: [21].) The survey involved over 80 planners and architects, some leading in their field, such as Raymond Unwin, Ernest Newton, and H. V. Lanchester. It delivered more than 300 diagrams and maps, which were exhibited in the Galleries of the Royal Institute of British Architects in 1920, attracting a good deal of attention among architects and planners [21]. Based on the maps produced (see: London Metropolitan Archives, LAM/4694), we may infer that the survey delivered utilization maps (i.e., land use maps) in their simplest form, in the sense that they only indicate the predominant use of spatial units that sometimes correspond to plots, sometimes to a set of plots of similar use, and sometimes to building blocks or even larger areas. A classification of uses was developed specifically for the Survey, organizing all uses into five broad categories (see Figure 1 in the article published by Hewitt in 2012 [21]). By today's standards, this would not be a well-developed classification scheme (special uses co-exist with broad categories, some categories have unclear functional identity, while the list of uses is not exhaustive), but the idea of developing a classification scheme to serve and homogenize all surveys sets a methodological milestone.

In the United States, the need for land use surveys was expressed by Katherine McNamara and Theodora K. Hubbard [13] (p. 114), and especially by Harland Bartholomew. Bartholomew, who was recruited as the nation's first full-time public-sector city planner in 1914, devised a nationally influential pack of techniques based on the systematic inventory of land uses (see Figure 5). During the next decades, he applied this pack to more than 500 plans for cities, counties, regions, and states [31]. Bartholomew is criticized for using land use survey methods to pursue an agenda of racial segregation and exclusion in U.S. cities [32] (p. 50); however, the national professional culture was grounded in this system of urban knowledge production, which governed city planning practice through much of the 20th century [31]. Irrespective of the spoken or unspoken intentions they served at the time, land use surveys quickly became a prerequisite of comprehensive planning, and by the 1930s, it became a *"customary procedure for city planners to prepare maps showing the existing uses of all property within the municipal area"* [33] (p. 10).

During the same period, large-scale land use surveys took place in the United States. These were initiated as work-relief programs, as had been the case in England two decades earlier. Specifically, during the Great Depression and under New Deal work-relief programs, spanning from 1933 to 1943, millions of able-bodied unemployed workers were put to work on government projects [34,35]. Under these programs, and for the first time in American history, land use surveys of this scale were conducted. The Chicago Land Use Survey, for example, employed about 10,000 people during its operation and covered over 20,000 city blocks and 212 square miles of urban area [36] (pp. 239–240). Similarly, the Los Angeles Land Use Survey covered approximately 450 square miles within the city boundaries and produced 347 land use maps (see Figure 6).

The main cartographic choices for the aforesaid maps were more or less similar and remained unvarying throughout the whole cartographic series. These choices were as follows: (a) the plot was chosen as the basic spatial reference unit for the identification of the predominant use(s), (b) the uses were organized into 12 categories, and (c) only up to two predominant uses per plot were mapped. Of comparable quality and organizational logic are the maps drawn up during the same period in Great Britain, such as the one included in the study *Preliminary Draft Proposals for Post-War Reconstruction in the City of London* [37] (see Figure 7). In this map, (a) the building was selected as the basic spatial reference unit, (b) the uses were organized into nine categories, and (c) only one predominant use was mapped. The choice to map only one predominant use per building is highlighted in the legend of the corresponding map, along with additional cartographic conventions and choices. Additionally, the map indicates the predominant use of larger geographic zones (see capital wording in golden font, very faintly placed above areas with undefined demarcation).

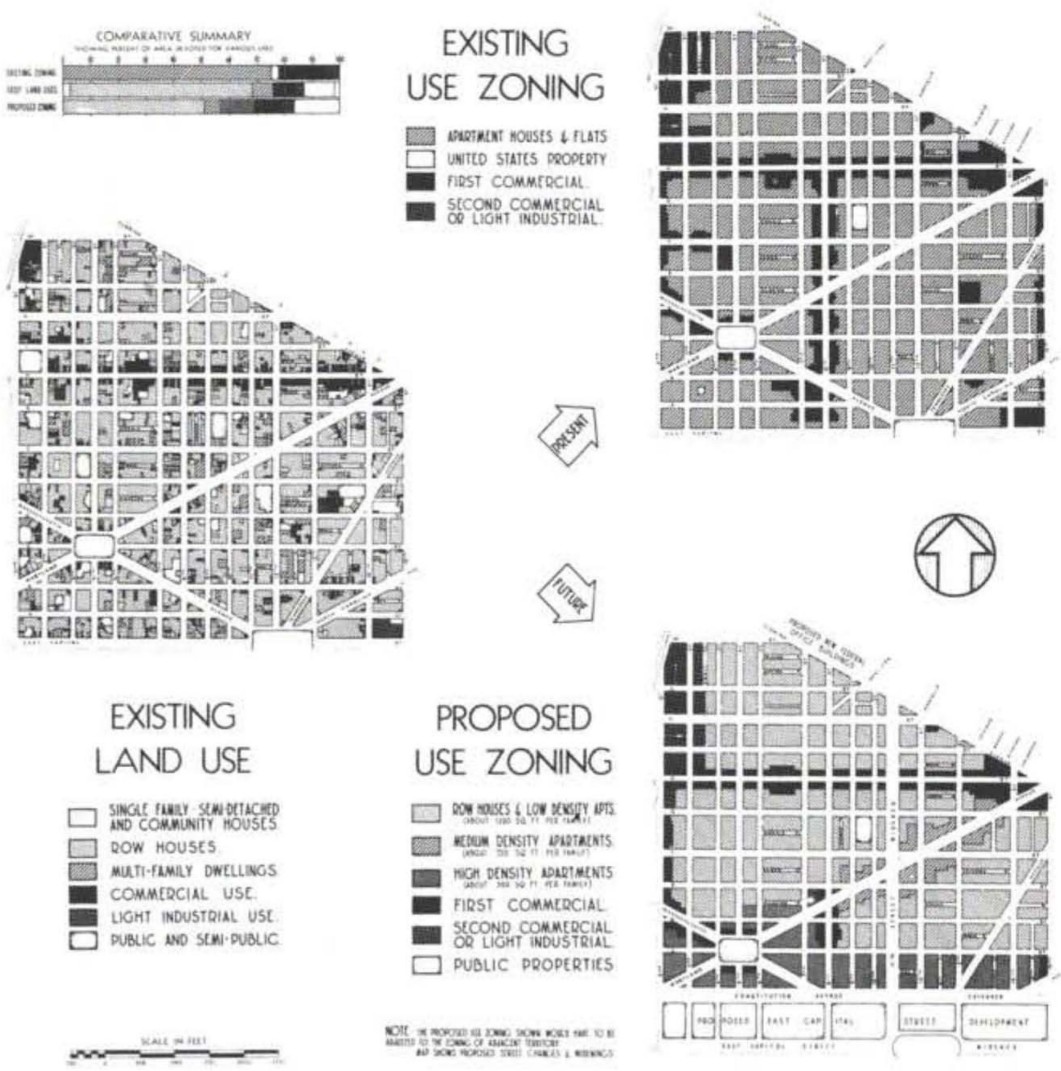

**Figure 5.** Diagram representing existing land uses and existing and proposed zoning for an area in northeast Washington, along with their comparative summary in the form of a bar chart (see upper left). The diagram is characteristic of the land use techniques widely utilized by Harland Bartholomew. Source: Land Uses in American Cities by Harland Bartholomew, assisted by Jack Wood, Cambridge, Mass.: Harvard University Press, Copyright © 1965 by the President and Fellows of Harvard College. Used with permission. All rights reserved.

The aforementioned surveys, especially when compared to the *Civic Survey of Greater London*, highlight the significant development in the practice of urban land use surveying and mapping carried out in these first decades. They also document the transition from empiricism to better-structured and more sophisticated approaches to land use surveying and mapping. This transformation should rather be attributed to the key contribution of practice, the needs of which were the main driving force for methodological development. Indeed, between 1935 and 1938, a number of handbooks were produced and distributed to engineers and surveyors [38,39] to assist the organization and standardization of the surveying procedure. The most notable one was the manual *A Technique for the Study of Land*

*Use in Cities and the Rural-Urban Fringe*, published by the Works Progress Administration in 1941 [40].

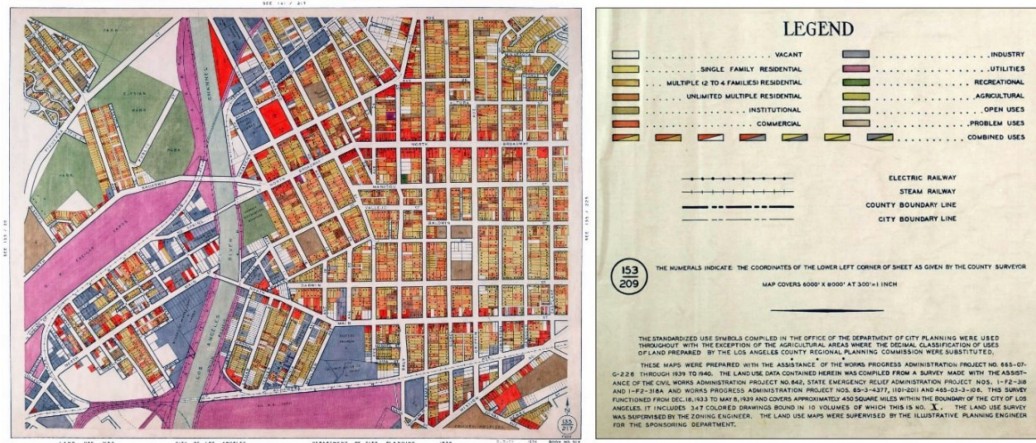

**Figure 6.** Land use survey map in Boyle Heights, Los Angeles. The survey covered approximately 450 square miles within the boundary of the city of Los Angeles. It included 347 colored drawings bound in 10 volumes, of which this is the 22nd map of the 6th volume. Source: USC Digital Library, http://doi.org/10.25549/wpamaps-m190 (accessed on 4 February 2022).

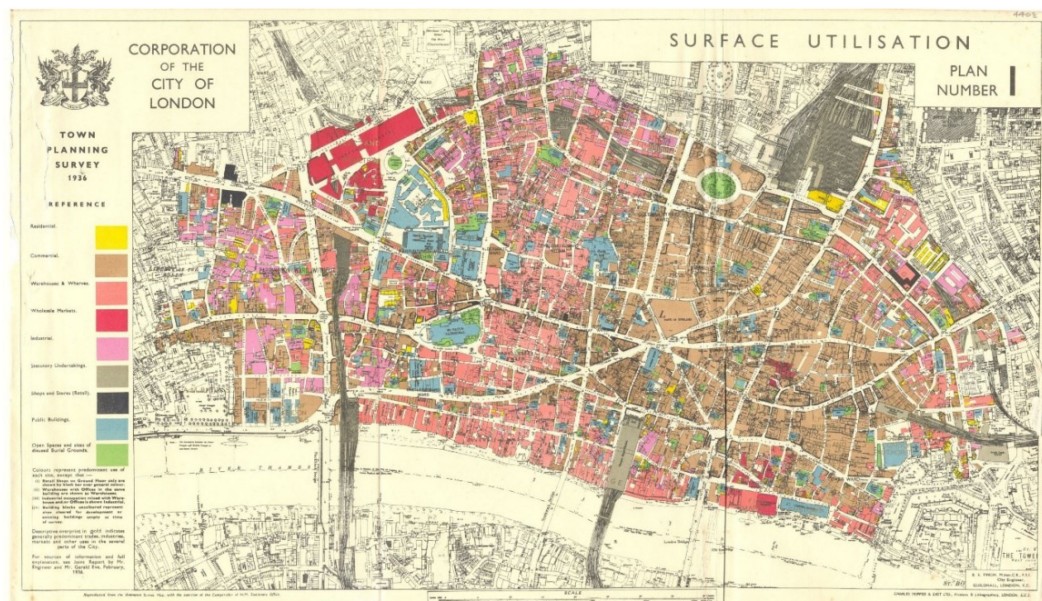

**Figure 7.** Land use survey of the City of London, 1936. Source: [37]. Available at http://www. antiquemapsandprints.com Copyright © Antiquemapsandprints.com. Used with permission. All rights reserved.

Following these, a considerable number of articles and critical reviews on land use survey methods were published (see: [38,41–44]). The methodological development is particularly evident in two articles, authored in 1941 and 1942 by Hugh Young, Chief Engineer of the Chicago Plan Commission, and Robert Filley, Technical Director of the Chicago Land Use Survey [41,42]. These articles summarize the knowledge produced up until then through Works Progress Administration surveys. For the first time, there was clear separation and systematic presentation of (a) the objectives of a survey, (b) the various spatial data that can be collected during a survey, (c) the different survey techniques, (d) the ways to process and analyze land uses, (e) the types of maps that can be produced, and (f) the ways to capitalize on the survey results, both in urban planning and in general.

Furthermore, the authors regarded urban surveys as an essential part of the planning process, a position which, within the specific historical context, indicated a significant degree of general acceptance of urban surveys as a pre-requisite of any planning practice.

Throughout the period of 1930–1950, additional large-scale land use surveys were planned and carried out. Although not specific to urban space, these surveys enhanced the overall methodological and administrative capacity of the sectors involved to carry out such extensive surveys [30]. Among them, a few stand out: the detailed survey of the use of every field in England, Scotland, and Wales in the early 1930s, organized and summarized in the final volume of *The Land of Britain: Its Use and Misuse* by Dudley Stamp [45], published in 1948; and *The World Land Use Survey*, carried out by the International Geographical Unit in 1949, with the financial support of UNESCO (in reference to those two and other important surveys, as well as to their role in the advancement of survey methods and techniques, see: [20,30,46–50]).

Soon, all this activity on surveys found its way into legislation, both in England and in the United States. In England, Section 5 of the 1947 Town and Country Planning Act stated that "*every local planning authority shall carry out a survey of their area, and shall [ . . . ] submit to the Minister a report of the survey together with the plan*". Since surveys were made compulsory for the preparation of any plan, the Ministry of Town and Country Planning prepared and published handbooks on survey techniques [51,52], so as to assist planners in this specific task (see also: [53] (pp. 139–140), [54] (pp. 88, 153–155)). Similarly, in the United States, the conduct of "*comprehensive surveys and studies of present conditions*" was instituted with the Standard City Planning Enabling Act (Advisory Committee on City Planning 1928, Section 7) and, according to Bartholomew [55] (p. 17), knowledge of land use became more than a planning precept; it acquired legal significance. Handbooks on the surveying and mapping of land uses were also published in the United States by the Public Administration Service [56,57]. In fact, the momentum behind the survey of urban space became so strong that it surpassed the interests of the professionals involved, to the point that school students in the United States were carrying out land use surveys as part of their typical education (see: [58,59]).

### 3. Survey Methods in the Spotlight of Academic Research

All this professional activity in land uses surveys, stimulated by developments in planning legislation and in transportation studies (the latter requiring extensive land use surveys, see: [12], [60] (p. 117)), seems to have triggered corresponding academic research. Specialized studies and monographs, as well as extensive sections in planning textbooks, dealing exclusively with the methodology of urban surveys, were published. For example, Lewis Keeble's book [54], published in 1952, included five chapters on techniques and methods for the survey and representation of urban space (Chapters 5 to 10), as well as one section (Section 9.2) specifically focused on the survey and representation of land uses. It should be noted that Keeble's book was regarded as one of the major reference books on town planning and maintained this position for more than a decade [14,53]. A few years later, in 1963, John N. Jackson published a book specifically focusing on surveys for town and country planning, which covered the issue of urban surveys even more comprehensively and in detail, devoting a chapter specifically to land use and building surveys [61] (pp. 108–129). The annotated bibliography *Selected References on Land Use Inventory Methods*, authored by Robert A. Clark in 1969 summarizes the sheer volume of the relevant literature produced at that time [12].

In essence, most of the contributions discussed so far are collections of the accumulated empirical knowledge gradually gained through the conduct of urban surveys. However, a limited number of studies have moved away from empiricism and adopted a more scientific approach. This represents a major shift, marking the transition from one era to another.

This shift towards more scientific approaches is, of course, part of a wider turn towards positivism, most notably in urban and human geography in the work of Harold Mayer, Clyde Kohn, Peter Haggett, Richard Chorley, and David Harvey [62,63]. In the case of land

use survey methods, the contributions influenced by this new approach formulated the basic theoretical and methodological principles and contributed to the development and refinement of the relevant terminology. This was achieved through placing the entire survey process within a broader epistemological framework or through treating land use surveys as an autonomous field of research. Such contributions originated from two different starting points:

The first is the *Urban Land Use Planning* textbook, published by Francis Stuart Chapin in 1957 (2nd edition in 1965), in which he devoted two chapters to the survey of urban land uses [64]. In trying to assess his work, we may note that his endeavor promoted land use surveying theory and methodology in a number of ways:

(a)　Initially, he refined land use terminology (an issue that is still open to discussion and further enhancement; see: [4]) through defining '*urban activities*' as the functional and dynamic elements of urban space that take spatial expression in the form of '*land uses*'.

(b)　He placed the survey and analysis of urban activities into a systems-thinking framework, in which the city planner must examine and interpret the evolution of activity systems in order to predict their future form and, eventually, to plan them in terms of land uses. In addition, he envisaged a '*continuing inventory system*', which would serve as an '*advanced warning system*' for anticipating important changes in the locus of activity patterns (see: [2]) (pp. 221–226 and 255).

(c)　He merged the following into a single framework of understanding: the typology of activities, the existing behavior patterns (i.e., movements of people, goods, and services among the different types of activities), the resulting spatial patterns, and the study of their possible future form, as well as their planning (pp. 226–231).

(d)　He provided the methods to be used for the survey and study of each special type of activity (i.e., firms, institutions, and households) and their sub-classes (pp. 231–253).

(e)　He provided technical guidelines and prepared the planner for real-world land use surveying practice in relation to the following: the selection or/and preparation of reference/base maps (pp. 257–264); the classification of land uses in a manner which is meaningful in terms of the activity systems discussed above (pp. 271–283); the needed preparation of the planner in advance of the actual field survey (pp. 284–285); the techniques utilized in field surveys (pp. 285–291); and the mapping of land uses (pp. 292–298).

Even if the contributions under (d) and especially (e) do not correspond to a theoretical level of elaboration, it is precisely the fact that his work deals with both theoretical and methodological issues as well as with technical and practical aspects of land use surveys that makes his contribution so valuable even by today's standards.

The second starting point is the research carried out particularly on the classification of uses. This is a matter of exceptional operational importance, both for the survey and for the general study and analysis of urban space [65]. The first study on the topic of land classification dates back to the work of Lovejoy in 1925 [66]. Some important studies on the subject were also published in 1941 in a special issue on land classification of the *Journal of American Institute of Planners* (see third issue of the seventh volume). However, it was not until the late 1950s that the first concrete theoretical foundations were laid. Robert Sparks first emphasized the need to develop a classification scheme that would be consistent, comprehensive, and flexible, in order to make surveys and research on urban land uses comparable [67]. Irving Shapiro [68] criticized the lack of clear criteria in land use classifications, which tended to mix economic, legal, architectural, and functional characteristics in the same classification scheme. He also proceeded with recommendations for structuring a classification scheme based solely upon the *activities* being performed on each specific site (see also: [60] (p. 117)). Last but not least, he stressed that only such a classification scheme could be used for land use mapping purposes. Nevertheless, it was Albert Guttenberg [69] who succeeded in producing a series of classification schemes that addressed all of the above theoretical and practical concerns. Specifically, each scheme was based on a single dimension of the following land use characteristics: degree of site

development, type of building, actual use (i.e., land use activity), economic 'over-use', and activity characteristics (e.g., size, rhythm, range of influence, and material effect on the senses).

At the same time, land use classification was dynamically advancing on behalf of government agencies. In particular, the Land Classification Advisory Committee of the Detroit Metropolitan Area published in 1962 the *Land Use Classification Manual* [70]. This manual was designed to make a coherent description of land use structure possible at any level of detail needed to serve the purposes of each survey, be it a village or a metropolitan area (see Foreword and Preface of this manual). Unfortunately, the classification scheme provided by this manual did not avoid the pitfalls that Shapiro [68] and Guttenberg [69] recognized in various earlier classification schemes. However, this manual was—and still is—in many ways a milestone in land use survey methodology, as it provided, for the first time in urban studies, a coherent, systematic, structured, and very detailed classification of uses, together with general recommendations and guidelines on survey and mapping techniques.

In 1965, the Urban Renewal Administration, working with the Bureau of Public Works, published the *Standard Land Use Coding Manual: A Standard System for Identifying and Coding Land Use Activities* (SLUCM) [5], which set even higher scientific standards in urban land use classification. First, the classification stood on solid theoretical ground: (a) through using the land use activity characteristic to structure the entire land use classification tree, and (b) through distinguishing land use activities from economic activities (such as those identified by *Standard Industrial Classification* [71], published by the US Bureau of the Budget in 1957). Second, it was very detailed, including 772 different types of urban land uses, but it was also practical because the land uses were structured into four hierarchical levels that allowed an agency (a) to select the level of detail deemed most appropriate for the analysis and representation of land uses, and (b) to customize the classification tree to meet the needs of specific studies. It also explicitly stated the classification criteria utilized for the formation of each category. Third, it defined existing terms more accurately (e.g., the notion of land use activity) and introduced new useful terms in land use classification (e.g., parent vs. auxiliary activities). Fourth, it included guidelines which were supported with indicative illustrations. Overall, the SLUCM comprises the most comprehensive classification and inventory manual of land uses to date.

In most of the studies, manuals, and textbooks produced during this second period, the authors did not over-theorize on their subjects of interest and fully understood the commitments and necessities of practice. In addition, the substantial contribution of government agencies to the further development of land use survey methodology is indicative of the strong bond forged between the world of theory and the world of practice during this era, although a separation was to follow in the coming years.

## 4. The Exclusion of Survey Methodology from Academic Research

After the mid-1960s, planning theory proliferated. However, this proliferation was accompanied by a general and rapid decline in interest in urban land use survey methods, particularly on the part of academics [72]. For example, Brian McLoughlin [73] (p. 131) proposed to the planning community the use of the *Standard Industrial Classification* (SIC) [71] as a guide for land use typology. However, the SIC was already dated at that time, due to the publication of the SLUCM, which was a much better classification in many ways. In addition, he did not address the practical uses of this classification, nor did he delve into the survey methods that were discussed in both the SLUCM and Chapin's *Urban Land Use Planning*. Other prominent textbooks of this era, such as those by George Chadwick [74] and Andreas Faludi [75], or the more practical textbooks by John Ratcliffe [76] and Margaret Roberts [77], omitted the survey topic entirely.

Nevertheless, the new planning ideas of the 1960s questioned neither the usefulness of survey methods nor the importance of conducting land use surveys. For McLoughlin's systems view of planning, land use surveys provided a basis for a deeper understanding of how cities actually function [78] (p. 62). In Faludi's rational view, surveys are part of the

'survey-analysis-plan' method, the precursor to his new procedural approach [78] (p. 66). The fading interest in survey methods was further diminished by wider developments in urban planning legislation. For example, urban planning in the UK evolved from development plans with highly detailed land use maps to strategic planning documents after 1974 [78] (p. 63), [79] (p. 43).

In the 1980s and 1990s, the communicative approach to planning drew attention to the role of power and citizen participation in the formulation of plans. Central to the communicative approach was *communicative rationality*, which criticized any given and well-established approach, in order to promote new ways of doing things based on consensus [80] (p. 272), [81] (p. 184). This approach continued the abstractness of planning theory, for which there was already growing criticism [78] (pp. 63–64, 96 and 126), and showed steadily decreasing interest in incorporating methodological elements. These characteristics widen the gap between theory and practice [82], and research on land use survey methods was marginalized within academia.

In contrast to these developments in academic research, real-world urban planning practice continued to demand better inventory systems; thus, various professional governmental bodies kept working on the topic. However, the research was solely focused on the production of new classification schemes of land uses. Specifically, in the UK, the *National Land Use Classification* was developed during the early 1970s by a team drawn from central and local government with the aim of devising a standard land use classification for the new style development plans introduced by the Town and County Planning Act of 1968 [83] (p. 9). A few years later, in the early 1980s, the Department of the Environment simplified this classification scheme for its own needs and renamed it to *Land Use Change Statistics* [84]. The latter served as the basis for the *National Land Use Database* that was published in the early 1990s. This new classification was prepared by the Office of the Deputy Prime Minister to address the various inconsistencies in land use definitions and categories, as well as to provide a consistent and complete land use classification scheme at the national level. Meanwhile, in the United States, the American Planning Association led the update of the SLUCM in the mid-1990s, which was still in use by planners (APA 1994). The new classification was released in 2000 [85]. Being a multidimensional and multiscale hierarchical land use classification model, it allowed users to have precise control over land use classifications [3] (p. 284).

Of course, it should be noted that the above new classifications were focused exclusively on the issue of identification and organization of uses into groups, completely ignoring issues related to the actual purpose for which they were prepared, i.e., for land use surveying. For example, none of these classifications provided guidance on how to prepare a survey, how to conduct a survey in the field, or how to map the surveyed uses. In contrast, all of the older classification manuals, especially the 1962 *Land Use Classification Manual* and the 1965 *SLUCM*, addressed these issues. This oversight is compounded by the fact that contemporary literature also fails to address these issues. This is true even of the most specialized textbooks, such as *Site Analysis: A Contextual Approach to Sustainable Land Planning and Site Design* by James LaGro, published in 2008 [86], or the *Site Planning and Design Handbook* by Thomas Russ, published in 2009 [87]. Even in the most recent fifth edition of *Urban Land Use Planning*, of which Chapin is no longer an author, the urban land use survey chapters have been removed (see: [88]). There are, however, some handbooks, such as the *Land Use Resource Guide: A Guide to Preparing the Land Use Element of a Local Comprehensive Plan*, published by the Center for Land Use Education in 2005 [89], or Gerrit Schwalbach's *Urban Analysis*, published in 2009 [90], but even these offer a limited and somehow simplistic overview of the survey task.

In addition, it is surprising to note that recent technological advancements had little impact on the survey methods of urban land uses for town planning purposes, although GIS has provided insightful tools for the spatial analysis of urban uses and urban land uses [91], while its integration with remote sensing has reshaped the methods for the survey of *land cover* [92,93]. Specifically, the blooming technology in remote sensing and GIS,

including the increased availability of accurate and comprehensive spatial data from various geo-platforms and the rapid development of hardware technology (in mobile devices, in unmanned aerial vehicles, and in satellite constellations), did not influence the field [6]. Though, a number of general-purpose field-mapping applications have been recently developed (e.g., ArcGIS Collector, Mapit Spatial, QField), which increase the convenience of the surveyor and ensure GIS interoperability, they have not contributed to the further development of the methodological part of the survey. These field-mapping applications mainly replicate the conventional (paper-based) inventory procedure conducted with field listing forms (see: [94]). Also, none of these applications have been particularly designed for the survey of urban land uses (see: [95]). Thus, their utilization requires customization, which compromises the gains in case of short-term use [96]. It should be clarified that the use of GIS generally offers increased convenience, when it comes to organizing, storing, and processing land use data, and constructing land use maps. But, again, no actual *methodological* gain is identified in the survey procedure.

## 5. Discussion

The preceding review of Anglo-American literature contributes to the discussion of land use survey methodology in three distinct ways.

First, it serves a very practical purpose, as it highlights the key readings in the field for anyone with a theoretical or practical interest in urban land use surveys. This is particularly important for those interested, since most of these readings were published in the 1950s and 1960s and cannot be easily traced today.

Second, it offers insights into the way(s) in which survey methods have evolved to date. These insights have led to the proposed periodization of this evolution in three phases, a contribution that enriches discussion and research on the history of planning methodology. Specifically, the first phase, which began at the end of the 19th century and stretched to the late 1940s, was characterized by two processes. The first relates to the general exploration and understanding of the necessity to conduct surveys prior to planning, in which Geddes' work was of key importance. Although this phase began through exploring the above necessity, it ends in the late 1940s with the international acceptance of surveys as an integral part of planning methodology. The second process relates to the connections between practice and methodological research. Namely, the initiation of the first large-scale urban land use surveys in the UK and the USA, which were considerably labor-intensive and required the systematization of methods and techniques, as well as a significant degree of coordination and administrative organization of the parties involved, were the trigger and the driving force behind the further development of the survey methods. This development is witnessed both in the qualitative transformation of the actual land use survey and mapping techniques, as well as in the production of the first specialized studies on survey methods. The land use survey was, for first time, established as a key and independent part of broader planning research.

The second phase, which starts in the 1950s, is characterized by a scientific approach to the issue of surveying land uses. This approach was triggered by the utilization of systems theory as a framework theory for land use analysis and planning, as well as by important advances in the classification of land uses. Of course, the utilization of scientific approaches to land use surveys should be assessed alongside similar trends in the related fields of urban and human geography. This phase is also characterized by the substantial contribution of government agencies to the further development of land use survey methods, which leaves no room to question the very close bond between theory and practice throughout the second phase.

On the contrary, the third phase, which starts in the 1970s, is characterized by a complete disruption of this relationship, as a result of the theory shifting towards abstraction and its increasingly lesser interest in dealing with and incorporating methodological elements. This third phase extends to this day and is further characterized by loss of

knowledge and a return to simplistic and empirical approaches to the issue of land use surveying, as is evident in the review of the contemporary literature.

The third contribution of the present study is that it allows us to detect the main areas of inquiry comprising the core of the specific field, as well as to identify the discussions developed in its periphery. Such an endeavor is subjective to the extent that different viewpoints and wider or narrower perspectives can be utilized for this identification, but it is significant because it structures the field and puts the various discussions in context.

Regarding the core of the field, at least two major areas of inquiry appear to belong there. The first one is substantiated as a set of *discussions on certain theoretical topics*. These discussions are not always explicit and, indeed, sometimes are manifested in the bibliography as tacit assumptions. One such silent discussion develops on the very *definition of the field*, with narrower or wider perspectives leading to different comprehensions of its boundaries. The narrow perspective, very much evident in contemporary readings, equates land use survey methodology with methods and techniques used in field inventories. The wider perspective includes all the theoretical and practical aspects of the whole process, which starts with the planning of a survey and concludes with the delivery of land use maps. Another discussion belonging here regards the *definition of the terms land use, urban use,* and *urban activity*. In many readings, especially the most contemporary ones, the use of these terms is problematic, as they are used interchangeably with their meaning varying according to context, while more often than not no (clear) definition is provided for each term. A recent article tried to shed some light on this discussion [4]; however, the field is still far from achieving widespread acceptance of the proposed terminology, as well as an understanding of the different cultural and scientific contexts that influence the use of these terms, especially between the United States and continental Europe. The classification of land uses also falls under the category of theoretical discussions. This appears to be, diachronically, the most heated debate, probably for two reasons: first, because land use classifications comprise a nodal part of planning legislation and, second, because classifications need to be revised periodically, since they are a product of their time, and to be adjusted to the specific objectives of each survey and study.

The second major area of inquiry concerns the *methodological part* of a land use survey. Based on the literature review, the discussions comprising this area can be organized under the following four topics, which correspond to the basic methodological steps of a land use survey. (a) *Planning of the survey*: this topic includes all discussions on the objectives, the resources, the timeframes, and the expected outcomes of a land use survey, as well as discussions on the identification of the exact land use properties that need to be surveyed and in relation to the practical obstacles in their collection. (b) *Preparation of field inventories*: this topic includes discussions on the technical and administrative preparative tasks required for a smooth and productive survey, which range from the production of base maps to training the surveyors. (c) *Conducting field inventories*: this includes all discussions on the methods and techniques utilized in field inventories, such as the use of field listing forms, the use of digital applications, the rules followed for the required simplification of building geometry, or the techniques used in ascertaining the use of non-accessible buildings. (d) *Processing of records and representation of land uses*: this topic is dominated by discussions on the application of the principles of cartography and thematic mapping in the production of land use maps; recent discussions on GISs and databases could also be included here.

In addition, there are a number of discussions that constitute the *periphery of the field*. These discussions establish connections with other fields of knowledge, contribute to wider debates, or link land use surveys to other phases of the planning process. The identification and the grouping of these discussions in topics fall out of the scope of the present article since the literature review did not focus on these peripheral discussions. A few indicative examples of the latter have already been mentioned, such as discussions linking land use surveys to racial segregation or considerations of the divergence between theory and practice. The present study on the evolution of land use survey methods is also a

discussion located in the periphery. It is expected, though, that the proposed periodization, the highlighting of key readings, the documentation of the insufficiency of the current literature, the structuring of the field according to areas of inquiry and topics, and the identification of the basic methodological steps will prompt additional research in core and peripheral discussions, to enhance the methods used for the survey of urban land uses.

**Funding:** This research received no external funding.

**Data Availability Statement:** No new data were created.

**Conflicts of Interest:** The author declares no conflict of interest.

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
