# Peer review of "Urban Land Use Survey Methods: A Discussion on Their Evolution"

_urbansci, doi:10.3390/urbansci7030076_

Round 1

Reviewer 1 Report

The title of the paper is: Urban land use survey methodologies: key contributions and discussion on their evolution.

The main purpose of this paper was to review the relevant Anglo-American literature on survey methodology published from the early 20th century to the present, and to highlight its key contributions.

The keywords are chosen accordingly.

The short introduction should highlights why the study is so important. The author rightly explain what the term 'survery' means in light of this article. Likewise, they rightly emphasize the meaning and difference in the terms 'urban use' and 'urban land use'. But in my opinion, these two figures (Fig 2 and 3) should be described a little differently. Because they relate to the phrase of 'urban land use', not to 'urban use' like in figure 1, then this phrase should be used.

Line 67:‘…how land is currently used…’ and what the dynamic of changes is - because land use is changing and methods have to be relevant to the time in which the changes are taking place.

Figure 1. under each figure is a description, e.g. 4A - block plan. These drawings should be marked 1A, 1B and 1C and under the drawing there should be an explanation of what 1A, 1B and 1C represent - then on line 161 you can refer to a specific drawing e.g. 1A. Because in the current description (line 160-161) it is not clear to which specific drawing this refers. Also, the quality of the drawings is poor which makes it difficult to locate this five broad categories of land uses.

The order of the chapters is correct.

In my scientific work, I constantly use GIS systems to research a new methodology of space analysis for specific purposes. Research connected with the different size of the base fields of the assessment of the space state it would take me months if GIS didn't exist - with which - I know the author will agree. However, the question is, whether these systems, allowing, for example, the use of model builder tools or similar, do not create a method of examining the state of space for specific purposes? e.g. analysis of space development to develop decision alternative map for some investments. The scale free networks theory as a new method in spatial planning the theory has been implemented. Also, the genetic algotirhms as a tool for supporting the processes of analysis and predicting urban development. The problem, in my opinion, is that less and less books are written that would summarize this knowledge in a reliable and up-to-date way.

I suggest that the author reconsider this opinion and observations.

Reviewer 2 Report

This paper reviews the history of land use surveying in the USA and UK in the past 125 years. Overall it is well written and clear, and provided an interesting read.

I am not familiar with the journal Urban Science and therefore the type of topics covered, but for me this article reads like book chapter rather than a journal paper. Although there is a discussion provided at the end, the focus here is mostly on bringing together a comprehensive history. It is not really explained why only the USA and UK are included, and they are not particularly examined in a comparative way. Largely the experience of the two countries tells a similar story, so the benefit of focussing solely on them is not clear. Perhaps more thought is needed here.

Minor edits:

Very notably the word methodology is used throughout when mostly meaning method. (Methodology is the study of methods rather than the carrying out of a method).

Line 7: ‘a fairly long tradition’. Vague. Re-state.

Line 54:’potentialities’. Replace with ‘potential’.

Lines 87-88: ‘they seem to concentrate’. Seem to or actually? Be specific here.

Line 105: what is a ‘figure-ground map’?

Lines 133 & 134: remove hyphens as a punctuation.

Line 172: ‘In our days’. Re-word this.

Line 285: ‘is the offspring of’. Replace.

Line 290: ‘in my view’. Delete.

Line 366: ‘In 1965 was published’. Re-write

Line 369-70: ‘upon which …. should be based on’. Re-write

Line 388: replace ‘divorce’ with separation.

Line 391: ‘hand-by-hand’. Replace

Line 497: ‘which I personally embrace’. Delete

Line 504: replace ‘we’.

Line 511: remove hyphens as a punctuation.

Reviewer 3 Report

This is an interesting paper, offering some key contributions on urban land use survey methodologies and their evolution. The approach and methodology presented would be very interesting for readers working in the associated field. The authors however should include a section on brief conclusions reflecting on general findings from the study. Otherwise, the discussions are also offering some useful insights. I do not have any further comments or suggestions for improvement of this paper. Excellent contribution to knowledge in the associated field.

Reviewer 4 Report

The manuscript titled “Urban land use survey methodologies: key contributions and discussion on their evolution” (Manuscript ID: urbansci-2375535) reviews the Anglo-American literature on land use survey methodology, published from the beginning of the 20th century to date, and highlights the key contributions. My main concerns and questions about the manuscript are:

11.  The author indicated that the article reviews the pertinent Anglo-American literature on survey methodology published from the beginning of the 20th century. I think the science is universal and therefore defining the scope as investigating only the Anglo American literature for urban land use analyses is not suitable.

22. I am not agree with the author about the recent technological advancements had little impact on the survey methodology of urban land uses. I think both remote sensing and GIS has great impact on urban land use studies in addition the technological development even change the survey types. Therefore the technology and land use survey relation should be analysed in more detail.

33. In the abstract it was indicated that the paper proposes the periodization of the methodological evolution in three phases. However, the three phases were not clearly stated in the manuscript.

44. As indicated in the journal web page “A single-blind review is applied, where authors' identities are known to reviewers” Therefore it is meaningless to not give the refered papers’ details by explaining it as “details removed for blind review purposes”.
